# Optical Coherence Tomography-Based Atlas of the Human Cochlear Hook Region

**DOI:** 10.3390/jcm12010238

**Published:** 2022-12-28

**Authors:** Lore Kerkhofs, Anastasiya Starovoyt, Jan Wouters, Tristan Putzeys, Nicolas Verhaert

**Affiliations:** 1Research Group Experimental Oto-Rhino-Laryngology, Department of Neurosciences, KU Leuven, 3000 Leuven, Belgium; 2Department of Neurosciences, Leuven Brain Institute, KU Leuven, 3000 Leuven, Belgium; 3Laboratory for Soft Matter and Biophysics, Department of Physics and Astronomy, KU Leuven, 3000 Leuven, Belgium; 4Department of Otorhinolaryngology, Head and Neck Surgery, University Hospitals of Leuven, 3000 Leuven, Belgium

**Keywords:** optical coherence tomography, human organ of Corti, intracochlear anatomy, cochlear implantation

## Abstract

Advancements in intracochlear diagnostics, as well as prosthetic and regenerative inner ear therapies, rely on a good understanding of cochlear microanatomy. The human cochlea is very small and deeply embedded within the densest skull bone, making nondestructive visualization of its internal microstructures extremely challenging. Current imaging techniques used in clinical practice, such as MRI and CT, fall short in their resolution to visualize important intracochlear landmarks, and histological analysis of the cochlea cannot be performed on living patients without compromising their hearing. Recently, optical coherence tomography (OCT) has been shown to be a promising tool for nondestructive micrometer resolution imaging of the mammalian inner ear. Various studies performed on human cadaveric tissue and living animals demonstrated the ability of OCT to visualize important cochlear microstructures (scalae, organ of Corti, spiral ligament, and osseous spiral lamina) at micrometer resolution. However, the interpretation of human intracochlear OCT images is non-trivial for researchers and clinicians who are not yet familiar with this novel technology. In this study, we present an atlas of intracochlear OCT images, which were acquired in a series of 7 fresh and 10 fresh-frozen human cadaveric cochleae through the round window membrane and describe the qualitative characteristics of visualized intracochlear structures. Likewise, we describe several intracochlear abnormalities, which could be detected with OCT and are relevant for clinical practice.

## 1. Introduction

Sensorineural hearing loss is one of the most common sensory deficits in the world and will affect up to 10% of the world population by 2050, according to the estimates of the World Health Organization [1]. Hearing disability has a large impact on the daily quality of life; it can lead to social isolation and even give rise to dementia, yet possibilities for effective treatment remain limited [2,3]. In the majority of cases, sensorineural hearing loss is due to structural damage of the cochlea, which is often treated with a cochlear implant in patients with severe hearing loss. Unfortunately, the surgical insertion of the stimulating electrode array often traumatizes the delicate intracochlear microstructures, which cannot be visualized intraoperatively. Targeted regenerative inner ear therapies are being researched, but their translation into clinical practice relies on the ability to diagnose the underlying pathology and administer the treatments to specific regions of the cochlea without collateral damage [4,5]. As such, the efficacy of current and future treatments largely depends on the ability to visualize the intracochlear anatomy.

The human cochlea is a very small (4 mm × 7 mm × 10 mm) and complex organ, deeply embedded inside the human skull bone, prohibiting direct visualization of its internal structure [6]. In current clinical practice, magnetic resonance imaging (MRI) or computed tomography (CT) is used for intracochlear diagnostics and preoperative planning. Unfortunately, these techniques fall short in imaging most intracochlear microstructures due to their limited resolution (0.5 mm) [6]. MicroCT and histology can provide high-resolution visualization of the internal cochlear anatomy, but these destructive methods are not viable for future (in vivo) clinical use [7,8].

Recently, the use of optical coherence tomography (OCT) in hearing research has been rising in both morphological and functional studies [9,10,11,12,13,14,15,16,17]. OCT is a nondestructive high-resolution imaging technique with similar working principles to ultrasonography. It uses low-coherence infrared light instead of sound waves, resulting in a higher micrometer-scale resolution. It was first applied in the field of ophthalmology, and due to its success, the use of OCT rapidly increased in other medical fields, such as oncology, cardiology, and dermatology [18,19,20]. 

A main advantage of OCT is the possibility to perform transmembrane OCT imaging, which happens through the round window membrane (RWM), providing the ability to image the first 1–3 mm of the most basal portion of the inner ear, namely the proximal hook region [14]. This eliminates the need to disrupt the cochlear homeostasis by opening it, which makes it a promising tool for atraumatic cochlear implantation, inner ear therapy, and diagnostics. In our previous work, we demonstrated that relevant structures such as the basilar membrane (BM), Osseous Spiral Lamina (OSL), Secondary Spiral Lamina (SSL), and spiral ligament (SL) can be identified in the proximal hook region [14]. However, the interpretation of human intracochlear OCT images is non-trivial for researchers and clinicians who are not yet familiar with this novel technology.

This study aims to extensively augment the knowledge of intracochlear structures by nondestructive transmembrane OCT imaging and to provide otological clinicians and researchers with an atlas of the human cochlear microanatomy. Likewise, we investigated whether abnormalities of the intracochlear structures can be visualized using OCT in fresh and fresh-frozen human temporal bones.

## 2. Materials and Methods

### 2.1. Sample Preparation 

Seventeen temporal bones were dissected and imaged with OCT in this study. All samples were harvested within 72 h post-mortem: 3 specimens were obtained from an anonymous donor at the Vesalius Institute of the University of Leuven; 14 specimens were retrieved from individuals who underwent a clinical brain autopsy at the University Hospitals of Leuven. Informed consent was obtained from all subjects, their next of kin, or legal guardian(s). No medical history or background information was known from the anonymous donors; only the age and the gender were known from the clinical brain autopsy donors. Harvesting and use of the temporal bones were conducted in accordance with the Helsinki declaration and approved by the Medical Ethics Committee of the University Hospitals of Leuven (S65502). To achieve maximal access to the RWM for transmembrane OCT imaging, the inner ears (approx. 10 mm × 10 mm × 20 mm) were dissected out of the temporal bones according to the previously described method by Starovoyt et al. 2019 [14]. The stapes was preserved to avoid leakage of the intracochlear fluid and entry of air bubbles into the cochlea. No drilling or blue-lining on the cochlea or labyrinthine complex was performed to maximize the structural preservation of the structures. After the surgical dissection, the isolated inner ears were thoroughly evaluated under a surgical microscope to exclude obvious anatomical malformations and trauma to the stapes, the RWM, the cochlear capsule, and the semicircular canals of the vestibular system. In addition, the stapes mobility and the integrity of the RWM were visually evaluated under the surgical microscope by confirming the movement of the RWM, without leakage of the intracochlear fluid, when gentle pressure is applied to the stapes. No pathological findings were detected during the inspection and dissection of the temporal bones. The cochleae were not fixed, dehydrated, or decalcified to avoid damaging the delicate intracochlear structures, such as the vulnerable sensory epithelium on the basilar membrane. 

### 2.2. OCT Imaging 

Extracted cochleae were imaged using a commercially available stationary spectral-domain OCT system (Telesto TEL220C1; Thorlabs, Lübeck, Germany), with an optical source having a broadband center wavelength of 1310 nm. The axial resolution of the system was either 4.1 or 5.5 µm for water and air, respectively. An objective lens was used with a working distance of 42.3 mm and an effective focal length of 54 mm (LSM04, Thorlabs), resulting in a lateral resolution of 20 µm. The number of A-lines was 1024, as well as the number of b-scans. Three-dimensional C-scans of the intracochlear structures were acquired through the RWM, whereby the scanning was performed at the rate of 10, 28, and 76 kHz without averaging. With a dimension of 512 × 512 × 1024 pixels for c-scans, the acquisition time was between 7 s (for 10 kHz) and 2 s (for 76 kHz). During imaging, the extracted cochleae were held in place using dental wax. They were viewed and analyzed in ThorImage software, version 5.1.3, whereby the signal range was adjusted to the dynamic range of each dataset. Highly reflective structures will appear as bright (in the grey scale) or red to red-white (in the false color scale) on the OCT images. The refractive index was set to 1.45, the average refractive index of the RWM, measured in two human cochleae according to the method of Tearney et al. [21]. 

For segmentation, a c-scan was saved as a stack of JPG files, which were then imported into Avizo (FEI Visualization Sciences Group and Thermo Fisher Scientific Inc., Bordeaux, France). In Avizo, the JPG files were manually segmented by experts in cochlear anatomy, who carefully evaluated the results. The voxel size for each dimension was also specified in Avizo.

### 2.3. Microcomputed Tomography (microCT)

The extracted cochleae were imaged after staining them in contrast for 72 h using a Phoenix Nanotom M MicroCT device (GE Measurement and Control Solutions, Germany) [22,23,24]. The device was equipped with a tungsten target and had a voxel size of 6.3 µm; it was operating at a voltage of 50 kV and a current of 531 µA, with an exposure time of 500 ms, without using a filter. Over 360°, 2400 frames were taken. The data were processed in Datos|x using scan optimization and exported as 16-bit .tiff slices, which were converted to .jpeg images.

### 2.4. Histological Preparations

One cochlea was sent for histological analysis. This sample was fixed in a 4% formaldehyde solution for 5 days, dehydrated in ethanol 50% and 70%, and imaged using standard microCT to guide the position of the 2D histological sections. LLS Rowiak (LaserLabSolutions, Hanover, Germany) performed polymethylmethacrylate embedding, OCT-guided sectioning with a laser microtome TissueSurgeon and staining of the slices with eosin-hematoxylin.

## 3. Results

Seven fresh and ten fresh-frozen were visually inspected by OCT in the three-dimensional plane. The fresh samples were analyzed and imaged within 24 h after the death of the donor, while the fresh-frozen samples were preserved by freezing them at −18° for a longer period of time and were thawed at 4° 24 h before OCT imaging was carried out. After visual inspection, fifteen samples were included for further analysis. Two samples were excluded because of the presence of air bubbles in the ST and SV. The sample characteristics are summarized in Appendix A. An overview of transmembrane OCT images of all cochleae is provided in Appendix B. When it comes to scanning rates, our results show that a lower scanning time (e.g., 76 kHz) results in a shorter acquisition time but a lower overall quality due to a decreased signal-to-noise ratio (SNR). On the other hand, a longer scanning time (e.g., 10 kHz) leads to higher overall quality because of an increased SNR, which enabled us to align structures within the sensory epithelium more clearly.

### 3.1. Intracochlear Microanatomy

First, we investigated intracochlear microanatomy by means of nondestructive transmembrane OCT imaging and studied their appearance on the OCT images. We were able to clearly distinguish previously described intracochlear structures. The RWM separates the inner ear from the middle ear and is the first structure to backscatter the infrared light; it has the brightest intensity in the OCT image. The segmentation of a 3D image in Figure 1A illustrates the relation of the intracochlear structures with respect to the RWM.

Underneath the RWM, the osseous spiral lamina (OSL) separates the scala tympani (ST) from the scala vestibuli (SV). This structure is recognizable on the one hand due to the gap between the vestibular (OSL-vp) and tympanic plate (OSL-tp), indicating the location of the auditory nerve fibers running in between, and on the other hand, due to the high reflectance and intensity of the bony structure, indicated in Figure 1B. The OSL-vp is located underneath the gap, where the nerves run through. The less reflective part underneath the OSL-vp is the tympanic lip of the limbus.

Laterally from OSL, the CPB can be recognized based on lower reflectivity than the OSL. Laterally from the CPB, the BM forms the base for the epithelial cells of the organ of Corti, appearing as a thin, mostly high-scattering layer.

Next to the BM towards the later wall, a triangle-shaped soft tissue structure is visible, the spiral ligament (SL). Besides its typical shape, the soft-tissue SL can be recognized based on its less reflective appearance on the OCT images. The most lateral part of the SL is often shadowed by the bony secondary spiral lamina (SSL) resting on top of the SL at the ST side. The SSL can usually be discerned well from the SL based on its higher reflectivity.

The Scala Media and Scala Vestibuli are separated by the Reissner’s membrane, visible on OCT images as a tilted thin layer running underneath the OSL, BM, and SL. The attachment of the Reissner’s membrane onto the OSL is often not clearly visible because it is covered by the shadow of the bony OSL. The visualized structures within the OCT image illustrated in Figure 1B, we compared using Masson-Goldner histological staining (Figure 1D, and on the other hand, using microCT (Figure 1E). In the figure, the microscopic view of the RWM was also added, where the blue arrow over the RWM corresponds to the visualized OCT cross-section, indicated by the blue arrow at the top of Figure 1B.

### 3.2. The Human Organ of Corti

Since the organ of Corti (OoC) is the most crucial part of the human hearing system and the target for various inner ear therapies, we studied its anatomy on OCT more in detail and added a close-up OCT image and additional schematic representation of the visualized structures in a fresh sample (Figure 2). The OCT images have the required resolution to delineate different parts within the OoC, where the tunnel of Corti (TC) and inner spiral sulcus (ISS) are operating as crucial landmarks. The ISS is covered by the tectorial membrane (TM), which separates it from the SM. Besides its typical shape, the TM can be recognized as a structure covering the medial part of the OoC. The TC, the ISS, and the TM form an important indication to estimate the location of the outer and inner groups of cells and can be followed through different cross-sections in a 3D image. The most lateral part of the TM covers the three OHC rows, which rest on top of the Deiter cells (DC) and are neighbored at the medial side by the outer pillar cells (OPC), all of which are located laterally from the TC. At the current resolution (4.73 µm × 4.62 µm × 2.39 µm), the individual cells cannot be distinguished from each other, and we indicated them on the OCT image as a group of ‘outer cells’. Likewise, we indicated the ‘inner cells’ (inner hair cells, inner phalangeal cells, and inner pillar cells) between the TC and the ISS based on the histological slice in Figure 2C.

At the modiolar side, the (thinner) TM is attached to the (thicker) spiral limbus, which can be distinguished from the OSL based on its lower reflectivity. In the proximal hook region, the spiral limbus sits on top of both the CPB and the OSL.

### 3.3. The Cochlear Partition Bridge and Spiral Limbus

Raufer et al. recently reported that in human cochleae, the OSL connects to the BM through an intermediary structure called the CPB, such that the OoC sits on top of the BM and the CPB [25]. OCT enabled us to study the microanatomy of the most proximal CPB in more detail (Figure 3). This structure could be clearly discerned on OCT images: the reflectivity of the soft-tissue CPB is lower than the bony OSL, and the CPB is thicker than the BM. Since the CPB is the shortest in the base, we were able to only image a small part of it in the proximate hook region. The spiral limbus is neighbored by both OSL and CPB and is visible as a less reflective structure, a soft tissue structure underneath the vestibular plate (Figure 3).

### 3.4. Deviating Appearance of Intracochlear Structures

Apart from the natural intracochlear anatomy, we investigated if transmembrane OCT also enables the detection of deviating appearance of the structures within the SM, relevant for intracochlear diagnostics. The most common deviation was the disappearance of the Organ of Corti OoC epithelium, leaving only the BM as a straight thin line between the CPB, OSL, and the SL, visible in Figure 4 (after freezing, false color and greyscale OCT), and Appendix B. In one case, transmembrane OCT imaging was performed in a fresh sample received a few hours after the death of the donor, thus before freezing and after freezing for six months at −18°. Before freezing, the sensory epithelium was present on top of the BM so that the combined central thickness amounted to 57 µm. After freezing, the epithelium could no longer be discerned, leaving the BM at a central thickness of 32 µm (Figure 4). The microscopic view on the RWM and Stapes in Figure 4 showed no clear differences before and after freezing. Overall, we saw more remnants of the sensory epithelium in fresh OCT-imaged samples compared to the samples, which were fresh-frozen and needed to be thawed before they were imaged for the first time. This may indicate freezing-induced damage.

Another non-natural appearance concerns the varying orientation and shape of the TM, visualized in Figure 5. Figure 5A illustrates the original position of the TM, covering the medial part of the sensory epithelium. In some samples, the TM was detached from the OoC, no longer forming a barrier between the ISS and SM and no longer covering the medial part of the sensory epithelium. This detachment was also often accompanied by deformation of the TM shape and closer proximity to the RM (Figure 5B,D). In Figure 5C, only the spiral limbus remains, and the TM appears to have disappeared.

Finally, the appearance of the RM varied between samples: in some cochleae, it formed a straight line, as shown in Figure 6A, whereas in the others, the contour of the RM was rather loose and irregular and oftentimes concave towards the SM (Figure 6B).

### 3.5. Factors Limiting Imaging through the RWM

In Figure 7, three samples were found to contain a pseudomembrane: one located at the superior rim, one at the anterior rim, and one covering the entire RWM. In Figure 7A,B, the pseudomembrane could be clearly discerned from the RWM as it was never fully attached to the RWM over its entire surface. In Figure 7C, the pseudomembrane covers the entire RWM, making it difficult to differentiate it from the RWM itself. However, it can be distinguished by the thin black line between the membranes, as shown in Appendix B. To clearly align the pseudomembrane covering the entire RWM, multiple cross-sections must be analyzed. On microscopic examination, the pseudomembrane covering the entire RWM appears as a white membrane, while the RWM is typically more transparent (Figure 7(1C)). In some samples, The presence of the pseudomembrane limited the penetration depth of the OCT into the cochlea, making the intracochlear structures less clear (Figure 7B).

## 4. Discussion

The objective of this study was to provide an OCT-based atlas of the human proximal hook region to aid future clinicians and researchers with the interpretation of transmembrane OCT images of the human cochlea. We performed transmembrane OCT imaging in a series of seventeen human cadaveric cochleae and analyzed the characteristics of the visualized intracochlear structures, which are relevant for inner ear diagnostics and therapies. Our results demonstrate that transmembrane OCT imaging enables nondestructive, high-resolution, 3D visualization of the intracochlear microstructures, which are crucial for hearing function, as well as the detection of abnormalities in these structures. To the best of our knowledge, this was also the first OCT imaging study performed on a large sample size of human cochleae.

We were able to consistently visualize important intracochlear structures in the proximal hook region through the RWM. The RWM is a crucial entry point to the cochlea for inner ear therapies, functional studies, and, most likely intracochlear diagnostics. The proximal hook region is characterized by complex anatomy and is critical for the implementation of inner ear therapies. In particular, the OSL, BM, SSL, and SL, which are structures of interest during cochlear implantation [26,27], could be visualized using transmembrane OCT imaging. Damage occurring at these structures during an electrode insertion can harm residual hearing, causing mechanical-induced hearing loss, inflammation, fibrosis, or ossification of the cochlea [27,28].

Pathologies of the inner ear inducing hearing loss mostly affect the OoC, making it a target structure for the development of gene and inner ear therapies and accurate diagnostics. Based on OCT-based visualization of important landmarks such as the TM, ISS, and TC, we could determine the position of the inner and outer hair cells together with the supporting epithelium surrounding them. At the current resolution, it was not possible to visualize the cells of the sensory epithelium individually; however, in the future, this can be tackled by using functional OCT [11,29,30,31]. On the other hand, improving the resolution could be achieved by increasing the spectral bandwidth of the light source and enhancing the resolution of the optical spectrometer that records the reflected light’s interference spectrum. However, spreading the signal over a larger detection array requires a higher optical output to maintain a similar signal-to-noise ratio (SNR). Some of these challenges have been addressed in a recent study, but these solutions have not yet been implemented in commercially available OCT systems [32].

Recently, the CPB, a soft tissue structure between the OSL and BM, has been identified in humans, differentiating it from the cochlear partition composition of rodents [25]. In this study, we were able to visualize the CPB in the proximal hook region using OCT imaging. Here, the CPB was a remarkably short structure, supporting the most lateral attachment of the spiral limbus. These observations are in line with the description of Raufer et al. [25], stating that the CPB becomes wider from base to apex in the human cochlea. The ability of transmembrane OCT to visualize CPB nondestructively in the proximal hook region of the human cochlea is highly promising for the functional study of inner ear mechanics [25].

In addition, we were able to detect intracochlear abnormalities with unprecedented detail using OCT in an intact human cochlea, which is highly relevant for intracochlear diagnostics and therapy. Out of seventeen specimens, the sensory epithelium was apparent in four fresh samples. In contrast to the other samples, these were not frozen before being imaged with OCT. In one sample, we observed normal sensory epithelium within a few hours post-mortem, but it disappeared after being frozen for six months, suggesting that freezing may have a destructive effect on the sensory epithelium. as it is remarkable that the sensory epithelium was mostly left in fresh samples, compared to fresh frozen samples. This is a significant finding, and on top of that, the sensory epithelium was most likely to be preserved in fresh samples compared to fresh-frozen samples. These findings highlight the importance of using fresh samples for reliable morphological and functional studies of the human cochlea and suggest that further research is needed on the effects of freezing and prolonged frozen storage on the quality of human temporal bones. Additionally, we also observed degenerated sensory epithelium in two fresh cochlear samples. We do not have any otologic background information for these samples, but these results might demonstrate the ability of transmembrane OCT to detect pathological changes in the human organ of Corti, which in clinical practice could be related to ototoxic drugs, noise exposure, aging, genetic factors, or other factors. Furthermore, OCT imaging could be a promising tool to monitor in vivo the efficacy of future regenerating therapies [33].

Other abnormalities were detected at the TM, which is believed to play a key role in hair cell activation in response to acoustical stimulation and undergo significant structural changes and degeneration with aging [34,35,36]. We were able to visualize detachment of the TM from the limbus, a structural indication that is potentially linked to (the onset of) age-related hearing loss and hence relevant for future studies regarding future inner ear (gene) therapies and nondestructive (preventive) diagnostics [35,36]. While no otologic background information about the donors is known, the median age of the donors was 77 (range between 55 and 90 years) and thus might show signs of age-related hearing loss. However, why hypothesize that aging will not significantly affect the location of the visualized intracochlear structures using OCT but further clinical application in hearing-impaired subjects would be necessary to investigate the effect of aging on the integrity of intracochlear structures.

Additionally, we were also able to evaluate the RM through the RWM. The RM separates the SM from the SV and normally is tense between its attachment at the OSL and the lateral wall. Included in our intracochlear OCT atlas are both a straight tensed RM and a flaccid RM. The cause of the latter appearance is not known, yet in certain pathologies such as Meniere’s disease, endolymphatic hydrops in the SM can cause distension of the RM [37]. In OCT images of the isolated cochlea, it is possible that the tension caused by the fluid in the scalae is altered, causing bulging of the RM.

Since OCT imaging was performed through the RWM, any overlying pseudomembrane was also imaged when present [38,39,40,41]. A pseudomembrane was noted in 18% of all samples, comparable to the findings of Sahin et al. [41]. Since the RWM is the main access point for transmembrane OCT imaging, the presence of a pseudomembrane could negatively affect the visualization of intracochlear structures. It is not yet clear what the cause of pseudomembranes might be and hence unpredictable whether it is present in a patient who qualifies for inner ear surgery and diagnostics [39]. Regarding inner ear therapy, a pseudomembrane could also negatively affect the diffusion of drugs administered by transtympanic injection into the cochlea [42].

Our results illustrate that transmembrane OCT imaging is a promising tool for clinical practice to nondestructively investigate the intracochlear anatomy in high resolution and in real time. The anatomy of the hook region is individually highly variable, and OCT can be crucial as a tool to anticipate the patient-specific anatomy, decreasing the risk for traumatic CI insertions [43,44,45]. With the rising interest in residual hearing preservation and atraumatic electrode insertion during CI surgery, understanding the round window area and the anatomy of the proximal hook region is of utmost importance [46]. Additionally, the hook region is also highly relevant regarding safe intratympanic injections, precise diagnostics, gene therapy, and in vivo studies of human intracochlear mechanics [47,48,49]. OCT meets the needed requirements for in vivo high-resolution imaging, with the advantage of in-depth transmembrane imaging, facilitating the anatomical investigation of the cochlear base without the need for opening the cochlea and disrupting its integrity [14,32,50,51].

OCT imaging does have certain limitations. First, we used extracted cochleae, providing the advantage of determining the optimal imaging angle, which might be unachievable in clinical settings. One solution might be using an endoscopic-based OCT through a transcanal approach, which might account for the limited degree of freedom one experiences within an entire skull [32,51]. To achieve comparable imaging results to a rigid OCT system, an endoscopic OCT system would need to be specially designed and optimized for imaging the cochlea. This might involve using a smaller, more flexible endoscope and a light source with a broader spectrum to improve the resolution and contrast of the images. It would also require advanced imaging algorithms and specialized image processing techniques to improve the overall quality of the images produced [32].

Furthermore, due to the limited imaging depth of the OCT, we were only able to image the approximate hook region. Because of this, the apex of the cochlea remains unreachable with OCT. In the future, fiberoptic tools might provide a solution for this, such as imaging of the intracochlear space during the insertion of a cochlear implant or during the injection of intracochlear therapies [50]. Finally, certain factors may negatively affect the quality of transmembrane OCT imaging: a thickened RWM, pseudomembrane, and remaining debris or fluid on top of the RWM. Lowering the scanning frequency, removing the debris, and aspiration of the fluid can overcome these limiting factors. In many cases, the pseudomembrane also can be carefully removed by the surgeon. The origin of a pseudomembrane is not yet known, but it might be related to middle ear infections. However, clinical evidence for this correlation is still lacking, and further research is needed to investigate this. A mucosal pseudomembrane found adjacent to the surface of the RWM may impact its permeability. They may either inhibit diffusion by acting as an additional barrier, protecting the RWM [41,52]. On the other hand, several pathologies, such as Menière’s disease or chronic otitis media, can cause the thickening of the RWM itself, which might negatively affect transmembrane OCT imaging and its future applications [53,54]. Additionally, the presence of blood in a clinical setting on top of the RWM could form an additional challenge, as blood is highly scattered. This could be helped with the aspiration of the blood drops and better control of the blood pressure, as it is currently conducted during cochlear implant surgery.

In summary, various studies investigated intracochlear anatomy using transmembrane OCT imaging; most of these were based on animal research [14,29,30,55,56,57]. Here, we provide an extensive imaging atlas of the human intracochlear anatomy in the proximal hook region, which would help future otologic researchers and clinicians to familiarize themselves with the features of intracochlear structures on OCT images. We were able to consistently visualize relevant intracochlear microstructures at a very high resolution and illustrated both normal and abnormal anatomical appearances using transmembrane OCT imaging. Being able to disentangle normal and abnormal composition of intracochlear structures, together with the fact that OCT is nondestructive and can be used in real-time in vivo, makes it a highly promising tool for clinical practice. OCT is a step forward towards aiding hearing-impaired patients in getting a safe insertion of inner ear therapies and enabling microstructural inner ear diagnostics, which is currently not possible.

## Figures and Tables

**Figure 1 jcm-12-00238-f001:**
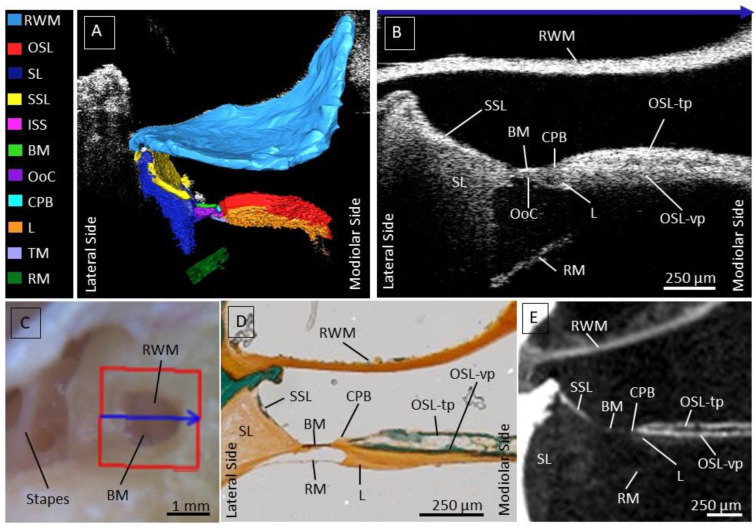
(**A**): Segmentation of a 3D OCT image; each structure is illustrated in a different color indicated in the legend. (**B**): OCT cross-section, with an a-scan rate of 10 kHz. Highly scattering structures result in whiter color, nicely indicating the border between the SSL and SL and CPB and OSL. (**C**): Corresponding microscopic view of the RWM, with the blue arrow indicating the location of the cross-section in (**B**). The location of the BM is visible through the transparent RWM as a dark line beneath the RWM. The stapes is located superior to the RWM. (**D**): Histology with Masson-Goldner staining as a reference to the structures on the OCT images. (**E**): Microcomputed topographical image, providing an extra comparison for the visualized structures on the OCT image in (**B**). Abbreviations: Round window membrane (RWM), Osseous spiral lamina—tympanic plate (OSL-tp), vestibular plate (OSL-vp), cochlear partition bridge (CPB), Basilar Membrane (BM), Organ of Corti (OoC), Secondary spiral lamina (SSL), Spiral ligament (SL), Spiral Limbus (L) and Reissner’s Membrane (RM).

**Figure 2 jcm-12-00238-f002:**
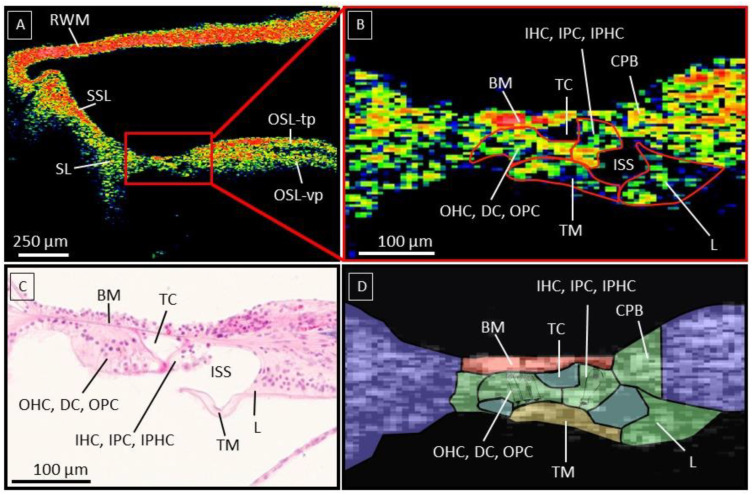
(**A**): illustrates an OCT cross-section (a-scan rate 10 kHz) of a fresh human temporal bone in color scale, with high scattering tissue in red. (**B**): is a zoomed-in cross-section of the OoC, with the cell groups indicated based on important landmarks such as the ISS, TM, and TC. (**C**): Histological section as validation for the OoC anatomy. (**D**): Schematic overview of the recognizable structures within the OoC. Abbreviations; Basilar membrane (BM), tunnel of Corti (TC), inner hair cells (IHC), outer hair cells (OHC), inner spiral sulcus (ISS), Tectorial membrane (TM), Spiral Limbus (L), Reticular Lamina (RL), Osseous Spiral Lamina tympanic plate (OSL-tp), OSL vestibular plate (OSL-vp), Reissner’s membrane (RM), Spiral Ligament (SL), Secondary Spiral Lamina (SSL), Round window membrane (RWM). Figure 2C was used with the written permission of the Massachusetts Eye and Ear otopathology temporal bone atlas.

**Figure 3 jcm-12-00238-f003:**
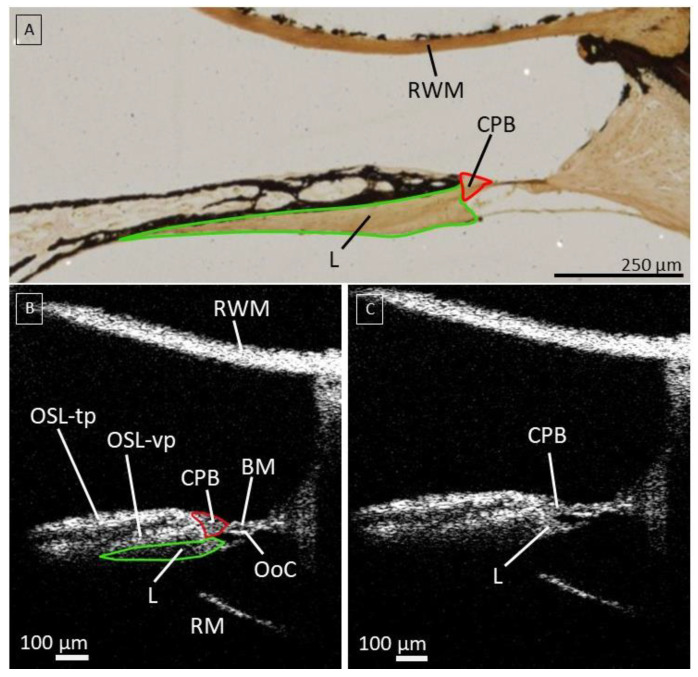
(**A**): Histological slice using von Kossa/von Gieson staining to illustrate the location of the CPB and L. (**B**); The cochlear partition bridge (CPB), aligned in red, is visible as a less reflective structure compared to the osseous spiral lamina illustrated in image (**C**). It is narrow at the base of the cochlea, indicated in the red circle in (**B**). The spiral limbus is aligned in green. (**C**) Same picture without labels and markings. A-scan rate of 10 kHz; Abbreviations: Round window membrane (RWM), Osseous spiral lamina—tympanic plate (OSL-tp), vestibular plate (OSL-vp), cochlear partition bridge (CPB), Basilar Membrane (BM), Organ of Corti (OoC), Spiral Limbus (L) and Reissner’s Membrane (RM).

**Figure 4 jcm-12-00238-f004:**
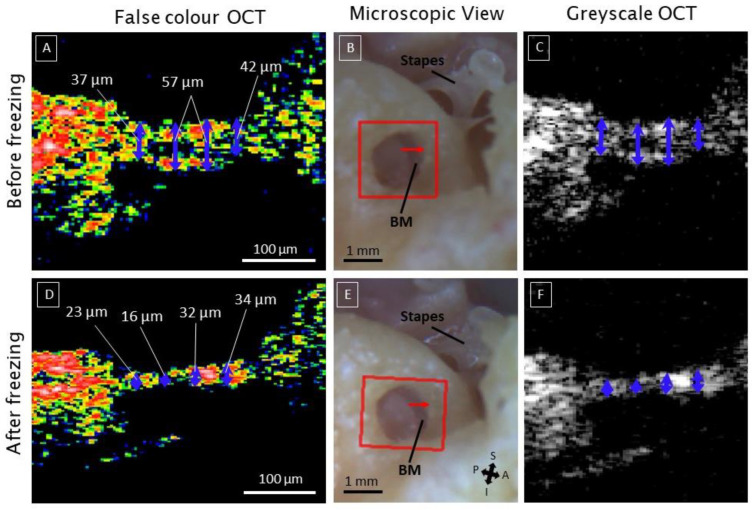
The disappearance of the hearing epithelium in the organ of Corti after freezing. This example is from the same sample at different scanning times. (**A**): the sample was first scanned after the release from the mortuary, and important landmarks within the OoC are visible, such as the ISS and TC. (**B**,**C**) are the corresponding microscopic and greyscale images, respectively. The red arrow indicates the location of the cross-section of the OCT images. (**D**): shows the same sample but at a later scanning date, where it was frozen and thawed again before the scanning. (**E**,**F**) are the corresponding microscopic and greyscale images, respectively. The blue arrows and corresponding thickness measures show how the freezing process altered the condition of the sensory epithelium. The a-scan rate of the OCT images was set to 76 kHz. Abbreviations; basilar membrane (BM), superior (S), inferior (I), anterior (A), and posterior (P).

**Figure 5 jcm-12-00238-f005:**
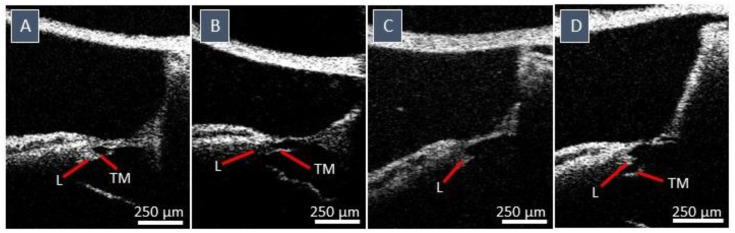
Different shapes, sizes, and orientations of the tectorial membrane (TM). (**A**): the TM in the original position, covering the medial part of the sensory epithelium. (**B**); clear distortions in the shape of the TM. (**C**): the TM disappeared. (**D**): the TM altered towards the SM; however, it must not be confused with the spiral limbus, which got loosened in these images. A-scan rate: 76 kHz Abbreviations: Tympanic membrane (TM) and spiral limbus (L).

**Figure 6 jcm-12-00238-f006:**
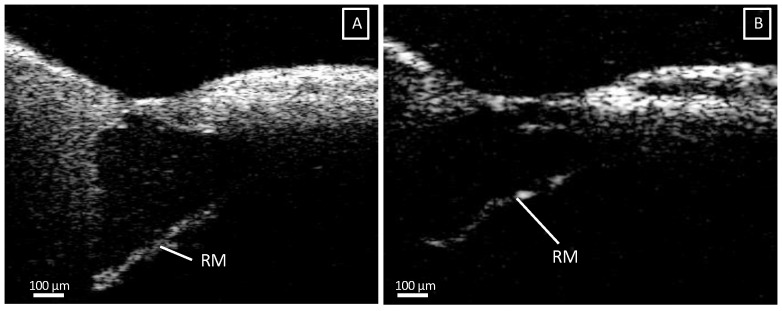
(**A**): Illustration of a straight Reissner’s membrane (RM), compared to an irregular, slightly concave towards the scala media (**B**). OCT images were taken with an a-scan rate of 76 kHz. Abbreviations: Reissner’s membrane (RM).

**Figure 7 jcm-12-00238-f007:**
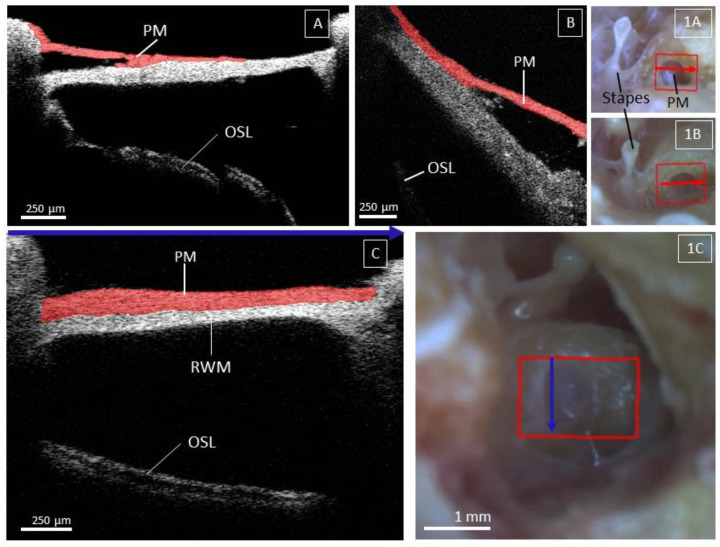
Visualization of pseudomembrane (PM) on top of the round window membrane (RWM). Figure (**1A**,**1B**,**1C**) are the corresponding microscopic images, with the blue and red arrows corresponding to the cross-section of the OCT images, indicated by the blue arrow at the top of figure (**C**). The shape, orientation, and extent of the PM were different across all samples and are illustrated in the Figure. (**A**,**1A**) show a PM covering the RWM at the superior rim. (**B**,**1B**) show a PM at the anterior rim of the RW niche, limiting the view onto the underlying OSL. (**C**,**1C**) show a PM covering the entire RWM; due to the perpendicular axis of the internal structures towards the OCT beam, the OSL is still visible; however, it is less reflective. (**A**–**C**) have an a-scan rate of 10, 76, and 28 kHz, respectively. Abbreviations: Round window membrane (RWM), Pseudomembrane (PM), Osseous spiral lamina (OSL).

## Data Availability

Data available upon reasonable request.

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
