# Peer review of "Optical Coherence Tomography-Based Atlas of the Human Cochlear Hook Region"

_jcm, 2022, doi:10.3390/jcm12010238_

Round 1
Reviewer 1 Report
In the current manuscript the authors present an atlas of intracochlear optical coherence tomography (OCT) images using a commercially available OCT setup. Seven fresh and ten fresh-frozen human cadaveric cochlea were imaged and results of 15 of these samples were included into the current manuscript. Imaging was performed through the round window membrane after sample preparation.
The manuscript is well written, the methodologies and results are presented in a clear manner. However, we have several minor and major comments which could strengthen the article:
-In the introduction in Line 55 a reference is missing.
-Why is the transverse resolution varying between 3.5 - 5 μm if the working distance and the focal length stay the same?
-What were the effects of the different scanning times used (10, 28, 76 kHz)?
-The aim of this manuscript was to give researchers which are not familiar with OCT an atlas to better understand the gained images. The reviewer believes that for a comprehensive comparison and understanding more ground truth data are needed. It would strengthen the manuscript to perform histology and microCT analysis of most samples. Some of these results should further be included into the manuscript and the rest could be added into a Supplementary material.
-Throughout the manuscript nearly no descriptions to figures and/or sub-figures are given. As a reader it is difficult to follow the results.
-Figure 1 (D), which staining was used for this histology image?
-How was the 3D segmentation in Fig. 1 performed and the 2D segmentation in Fig. 2?
-Did the authors saw the freezing induced shrinkage only in one sample or in all?
-In general, for the deviated appearance of the intracochlear structures (also the pseudomembrane later on) a histology comparison is crucial. It could clarify if the deviations are caused by freezing effects only or if the patients had some medical conditions which can be detected by OCT. It would also enable a quantitative analysis of the size and thickness of various structures investigated using OCT, histology and microCT.
-In the discussion section the author describe that cellular resolution was not achieved. The authors could discuss the possibility of an improved resolution by using a light source with a broader spectrum.
-Concerning the age of the included patients: Is there a possibility that the observed results will change if intracochlear of younger patients are imaged? If yes, the described atlas might be misleading for clinicians?
-References are missing in in the discussion in Line 395.
-The authors discuss the limitations of OCT for in vivo human cochlea investigations. It would be interesting to extend this discussion to existing endoscopic OCT systems and evaluate if these setups could achieve comparable imaging results or which type of setup would be needed.
-Please double check all references, it seems in some cases crucial information is missing.
-Please double check if all related work has been cited such as:
https://www.nature.com/articles/s41598-021-95991-8
https://www.ncbi.nlm.nih.gov/pmc/articles/PMC6571188/
https://opg.optica.org/ol/fulltext.cfm?uri=ol-43-9-1966&id=385623
Author Response
Dear Reviewer,
Thank you for considering our revised manuscript and for providing us with a thorough and interesting review. We have addressed all comments from the Reviewers and believe this has resulted in an improved manuscript. Please find the point-by-point list of our corrections below. All answers are provided in red color and all changes can be tracked as comments in the revised manuscript, which has been added as a pdf file.
Yours Sincerely,
Lore Kerkhofs, Anastasiya Starovoyt & Nicolas Verhaert
Reviewer comments:
Reviewer #1:
Point 1: In the introduction in Line 55 a reference is missing.
These references have been added (line 56).:
- Brody, K. M., Hampson, A. J., Cho, H. jung, Johnson, P., & O’Leary, S. J. (2020). A new method for three-dimensional immunofluorescence study of the cochlea. <i>Hearing Research</i>, <i>392</i>. https://doi.org/10.1016/j.heares.2020.107956
- Montgomery, S. C., & Cox, B. C. (2016). Whole mount dissection and immunofluorescence of the adult mouse cochlea. Journal of Visualized Experiments, 2016(107). https://doi.org/10.3791/53561
Point 2: Why is the transverse resolution varying between 3.5 - 5 μm if the working distance and the focal length stay the same?
Thank you for pointing this out, we erroneously used the term ‘transverse’ resolution. It actually concerns the ‘axial’ resolution. Depending on the medium used to image through, the axial resolution in our system and used system settings is either 4.1 μm in water or 5.5 μm in air. We clarified this in the manuscript (lines 110-112, visible as track-changes - all markup).
Point 3: What were the effects of the different scanning times used (10, 28, 76 kHz)?
When it comes to scanning rates, our results show that a lower scanning time (e.g. 76 kHz) results in a shorter acquisition time but a lower overall quality due to a decreased signal-to-noise ratio (SNR). On the other hand, a larger scanning time (e.g. 10 kHz) leads to a longer acquisition time but higher overall quality because of an increased SNR. When a high scanning time is used (such as 10 kHz), we are able to see different structures within the sensory epithelium better due to an increased SNR (lines 153-158).
Point 4: The aim of this manuscript was to give researchers which are not familiar with OCT an atlas to better understand the gained images. The reviewer believes that for a comprehensive comparison and understanding more ground truth data are needed. It would strengthen the manuscript to perform histology and microCT analysis of most samples. Some of these results should further be included into the manuscript and the rest could be added into a Supplementary material.
We thank the reviewer for this comment. Indeed the aim is to provide, within the available constraints of an article, as much relevant information for clinicians and researchers, new and experienced in this topic, as possible. We, therefore, have built up several figures on OCT imaging. In Figure 1, we added an extra comparison, next to the already available OCT and clear histological image, using a microCT image (Fig. 1(E)). This figure is also added in appendix B, since the soft tissue might be hardly visible in a smaller figure.
In Figure 3, we added an extra histological comparison to illustrate the location and a ground truth comparison regarding the spiral limbus and bridge, visualized in our OCT atlas. The staining we used here is von Kossa/van Gieson. In the histological slice, the spiral limbus is indicated in green, while the CPB was indicated in red.
For the illustration of the pseudomembrane and in support of Figure 7 we added corresponding microCT images with a visible pseudomembrane in appendix B.
Additionally, we have added relevant uCT images and concomitant analysis in the SI to strengthen our claims and provide the reader with more ground truth comparisons. These microCT images are accurate enough to provide a comprehensive comparison. Because microCT data was originally not taken up in the manuscript, we added a brief paragraph in the methods section for this (lines 130-137, track changes – all markup: “The extracted cochleae were imaged after staining them in contrast for 72 hours using a Phoenix Nanotom M MicroCT device (GE Measurement and Control Solu-tions)[22–24]. The device was equipped with a tungsten target and had a voxel size of 6.3 µm, it was operating at a voltage of 50 kV and a current of 531 µA, with an expo-sure time of 500 ms, without using a filter. Over 360°, 2400 frames were taken. The da-ta was processed in Datos|x using scan optimization and exported as 16-bit .tiff slices, which were converted to .jpeg images”).
Point 5: Throughout the manuscript nearly no descriptions to figures and/or sub-figures are given. As a reader it is difficult to follow the results.
The required description of the figures has been added to the main text.
- Figure 1: Lines 171, 188-192. (“The visualized structures within the OCT image illustrated in Fig. 1(B), we compared using Masson-Goldner histological staining (Fig. 1(D), and on the other hand using microCT (Fig. 1(E)). In the figure, the microscopic view of the RWM was also added, where the blue arrow over the RWM corresponds to the visualized OCT cross-section, indicated by the blue arrow in the top of Fig. 1(B).”)
- Figure 2: Lines 206-207, 219-220 (“Since the organ of Corti (OoC) is the most crucial part of the human hearing sys-tem and the target for various inner ear therapies, we studied its anatomy on OCT more in detail and added a close-up OCT image and additional schematic representation of the visualized structures in a fresh sample (Fig. 2).”, “based on the histological slice in Fig. 2(C).”)
- Figure 3: Lines 239 (“OCT enabled us to study the microanatomy of the most proximal CPB more in detail (Fig. 3).”)
- Figure 4: Lines 262-265, 268-272: (“visible in Fig. 4 (after freezing, false colour and greyscale OCT), and appendix B. In one case, transmembrane OCT imaging was performed in a fresh sample received a few hours after the death of the donor, thus before freezing, and after the freezing for six months at -18°.”, “The microscopic view on the RWM and Stapes in Fig. 4, showed no clear differences before and after freezing. Overall, we saw more remnants of the sensory epithelium in fresh OCT-imaged samples, compared to the samples which were fresh-frozen and needed to be thawed before they were imaged for the first time. This may indicate freezing-induced damage.”)
- Figure 5: 274-275, 276-277, 287-279 (“visualized in Fig. 5. Fig. 5(A) illustrated the original position of the TM, covering the medial part of the sensory epithelium.”, In some samples, the TM was detached from the OoC, no longer forming a barrier be-tween the ISS and SM, and no longer covering the medial part of the sensory epitheli-um. This detachment was also often accompanied by deformation of the TM shape, and closer proximity to the RM (Fig. 5(B) and (D)). In Fig. 5(C), only the spiral limbus remains and the TM appears to have disappeared.”)
- Figure 6: 298, 299: (“as shown in Fig. 6(A), whereas in the others the contour of the RM was rather loose and irregular, and oftentimes concave towards the SM (Fig. 6(B)).”)
- Figure 7: 306-307, 309-315: (“In Fig. 7, three samples were found to contain a pseudomembrane: one located at the superior rim, one at the anterior rim, and one covering the entire RWM. In Fig. 7(A) and (B), the pseudomembrane could be clearly discerned from the RWM as it was never fully attached to the RWM over its entire surface. In Fig. 7(C), the pseudomem-brane covers the entire RWM, making it difficult to differentiate it from the RWM it-self. However, it can be distinguished by the thin black line between the membranes, as shown in Appendix B. To clearly align the pseudomembrane covering the entire RWM, multiple cross-sections must be analyzed. On microscopic examination, the pseudo-membrane covering the entire RWM, appears as a white membrane, while the RWM is typically more transparent (Fig. 7(C1)).”)
Point 6: Figure 1 (D), which staining was used for this histology image?
Masson-Goldner histological staining was used for the image in Fig. 1(D). (Line 187).
Point 7: How was the 3D segmentation in Fig. 1 performed and the 2D segmentation in Fig. 2?
In the methods, we added a paragraph where we briefly explained that we used Avizo for the 3D segmentation in Fig. 1 (lines 124-127: “For segmentation, a c-scan was saved as a stack of JPG files, which were then im-ported into Avizo (FEI Visualization Sciences Group and Thermo Fisher Scientific Inc). In Avizo, the JPG files were manually segmented by experts in cochlear anatomy, who carefully evaluated the results. The voxel size for each dimension was also specified in Avizo”). In Fig. 2 we added a visual overlay of colors to indicate the structures more clearly. Expert researchers in cochlear anatomy indicated, aligned, and examined the structures within the OoC, and as a result, Fig 2 is a schematic illustration.
Point 8: Did the authors saw the freezing-induced shrinkage only in one sample or in all?
The authors only observed this freezing-induced shrinkage in one sample, but the difference in the integrity of the sensory epithelium before and after shrinkage was significant. We only saw differences regarding soft tissue i.e. the sensory epithelium. Further research is necessary to investigate the effects of freezing on the cochlea, but we hypothesize that this influences the integrity of the sensory epithelium based on this finding in one sample. (Lines 262-272, 371-385 : “In contrast to the other samples, these were not frozen before being imaged with OCT. In one sample, we observed normal sensory epithelium within a few hours post-mortem, but it disappeared after being frozen for six months, suggesting that freezing may have a destructive effect on sensory epithelium. as it is remarkable that the sensory epithelium was mostly left in fresh samples, compared to fresh frozen samples. This is a significant finding and on top of that, the sensory epithelium was most likely to be preserved in fresh samples, compared to fresh-frozen samples. These findings highlight the importance of using fresh samples for reliable morphological and functional studies of the human cochlea, and suggest that further research is needed on the effects of freezing and prolonged frozen storage on the quality of human temporal bones. Additionally, we also observed degenerated sensory epithelium in two fresh cochlear samples. We do not have any otologic background information for these samples, but these results might demonstrate the ability of transmembrane OCT to de-tect pathological changes in the human organ of Corti, which in clinical practice could be related to ototoxic drugs, noise exposure, aging, genetic factors, or other factors.”)
Point 9: In general, for the deviated appearance of the intracochlear structures (also the pseudomembrane later on) a histology comparison is crucial. It could clarify if the deviations are caused by freezing effects only or if the patients had some medical conditions which can be detected by OCT. It would also enable a quantitative analysis of the size and thickness of various structures investigated using OCT, histology and microCT
- “In general, for the deviated appearance of the intracochlear structures (also the pseudomembrane later on) a histology comparison is crucial.”: Histology does not always correctly show the intracochlear anatomy, the inner ear is often damaged during the process of tissue preparation and sectioning, which can affect the accuracy of the histological analysis. Additionally, different histological techniques can produce different results, so the accuracy of the analysis may depend on the specific technique that is used., which can make it difficult to obtain reliable and consistent results. Hence, we are not able to provide the readers with a histology comparison of the deviated appearance of intracochlear structures.
- “It could clarify if the deviations are caused by freezing effects only or if the patients had some medical conditions which can be detected by OCT”. There were 6 fresh samples, which means that the sample was not frozen, so the anatomy could not be influenced by the freezing process. We did see that in the samples which were fresh, the sensory epithelium was more likely to be apparent compared to the integrity of the sensory epithelium in the fresh-frozen samples. We do not have any medical background information from the donors, per the outlines of the donation process and the approval of the ethical committee. We can only describe what we see on the OCT images, and illustrate that differences in the position of the TM or other deviated appearances can be detected, which would mean that using OCT is already a big step forward toward intracochlear diagnostics, therapy, or cochlear implant surgery. Further research is needed to make big statements about the medical condition of donors and the appearance of anatomical structures. (Lines 371-385).
Point 10: In the discussion section the author describes that cellular resolution was not achieved. The authors could discuss the possibility of an improved resolution by using a light source with a broader spectrum.
Increasing the spectral width of the light-source is one component to increasing the resolution, the other is to increase the resolution of the optical spectrometer that records the interference spectrum of the reflected light. As the signal is spread out over a larger detection array a higher optical output is required to maintain a similar SNR. Some of these issues have been tackled in a recent publication of Lyer et Al (https://www.nature.com/articles/s41598-021-95991-8) but are not found in commercially-available OCT systems yet. (Lines 352-358)
Point 11: Concerning the age of the included patients: Is there a possibility that the observed results will change if intracochlear of younger patients are imaged? If yes, the described atlas might be misleading for clinicians.
It is until to date not well understood which specific changes occur in the inner ear due to aging. We hypothesize that gross intracochlear structures such as the SSL, OSL, and SL can be affected by tissue degeneration, but the location of the structures stays the same, and in this aspect our imaging atlas provides a good example of different visible structures. It is presumably mainly the composition of the structures themselves or the neurological composition/transmission which could change.It is difficult to obtain samples of younger patients, hence we added a line in the discussion (line 396-399). Overall, structures will be visible and recognizable in the same manner in younger and older patients (e.g. the SSL and SL can still be aligned due to differences in scattering properties of the tissue these structures are composed of). The greatest variation is of course, suspected to be in the sensory epithelium because hair cells degenerate and die due to aging.
Point 12: References are missing in the discussion in Line 395.
The following references have been added:
- Starovoyt et al 2019 Starovoyt, A., Putzeys, T., Wouters, J., & Verhaert, N. (2019). High-resolution Imaging of the Human Cochlea through the Round Window by means of Optical Coherence Tomography. Scientific Reports, 9(1). https://doi.org/10.1038/s41598-019-50727-7
- Dong et al. 2018 Dong, W., Xia, A., Raphael, P. D., Puria, S., Applegate, B., & Oghalai, J. S. (2018). Organ of Corti vibration within the intact gerbil cochlea measured by volumetric optical coherence tomography and vibrometry. J Neuro-Physiol, 120, 2847–2857. https://doi.org/10.1152/jn.00702.2017.-There
- Cho et al. 2022 Cho, N. H., Wang, H., & Puria, S. (2022). Cochlear Fluid Spaces and Structures of the Gerbil High-Frequency Region Measured Using Optical Coherence Tomography (OCT). JARO - Journal of the Association for Research in Otolaryngology, 23(2), 195–211. https://doi.org/10.1007/s10162-022-00836-4
- Cooper et al. 2018 Cooper, N. P., Vavakou, A., & van der Heijden, M. (2018). Vibration hotspots reveal longitudinal funneling of sound-evoked motion in the mammalian cochlea. Nature Communications, 9(1). https://doi.org/10.1038/s41467-018-05483-z
- Kim et al. 2018 Kim, W., Kim, S., Oghalai, J. S., & Applegate, B. E. (2018). Endoscopic optical coherence tomography enables morphological and subnanometer vibratory imaging of the porcine cochlea through the round window. Optics Letters, 43(9), 1966. https://doi.org/10.1364/ol.43.001966
- Burwood et al. 2019 Burwood, G. W. S., Fridberger, A., Wang, R. K., & Nuttall, A. L. (2019). Revealing the morphology and function of the cochlea and middle ear with optical coherence tomography. In Quantitative Imaging in Medicine and Surgery (Vol. 9, Issue 5, pp. 858–881). AME Publishing Company. https://doi.org/10.21037/qims.2019.05.10
Point 13: The authors discuss the limitations of OCT for in vivo human cochlea investigations. It would be interesting to extend this discussion to existing endoscopic OCT systems and evaluate if these setups could achieve comparable imaging results or which type of setup would be needed.
To achieve comparable imaging results to a rigid OCT system, an endoscopic OCT system would need to be specially designed and optimized for imaging the cochlea. This might involve using a smaller, more flexible endoscope and a light source with a broader spectrum to improve the resolution and contrast of the images. It would also require advanced imaging algorithms and specialized image processing techniques to improve the overall quality of the images produced. This is because the cochlea is a small, delicate structure located deep within the ear, and it can be difficult to access with an endoscopic (Lines 435-440) + added reference https://www.nature.com/articles/s41598-021-95991-8
Point 14: Please double-check all references, it seems in some cases crucial information is missing.
The references have been checked and crucial information has been added.
Point 15: Please double check if all related work has been cited such as:
- https://www.nature.com/articles/s41598-021-95991-8
- [32], lines 358, 428, 434,439]
- https://www.ncbi.nlm.nih.gov/pmc/articles/PMC6571188/
- [57]; line 462
- https://opg.optica.org/ol/fulltext.cfm?uri=ol-43-9-1966&id=385623
- [56]: line 462
Thank you, the authors included these references

Reviewer 2 Report
Review of JCM 2089623
Title: Optical coherence tomography-based atlas of the human cochlear hook region
Authors: L. Kerkhofs et al.
This paper presents ex vivo OCT imaging of human cochlea, aimed at helping for surgeons or clinicians unfamiliar with OCT to interpret the cochlear microanatomy displayed in the OCT images. The paper is very well-written in English and the readability is good. The image quality is also quite good.
1. At line 56, please add the prior OCT angiography (OCTA) studies for cochlea vascular imaging.
2. Authors mentioned that the lateral resolution is 3.5~5 um. However, LSM04 lens used has a focal beam spot over 30 um in spec provided from the vendor, which means that the lateral resolution is more than 30 um.
3. For the acquired 3D OCT dataset, please explain the number of pixels/A-line, number of A-lines/B-scan, number of B-scans/C-scan, and the scanned area.
4. At line 130, what is difference between fresh and fresh-frozen?
5. Briefly, explain about the method for segmentation shown in Fig. 1A.
6. Indicate the scale bar in Fig. 2C.
7. At line 227, please explain why the epithelium is not seen after freezing.
8. At line 267, please explain how to figure out PM on the RWM in Fig. 7C, and the cause of PM generation.
Author Response
Dear Reviewer,
Thank you for considering our revised manuscript and for providing us with a thorough and interesting review. We have addressed all comments from the Reviewers and believe this has resulted in an improved manuscript. Please find the point-by-point list of our corrections below. All answers are provided in red color and all changes can be tracked as comments in the revised manuscript, which has been added as a pdf file.
Yours Sincerely,
Lore Kerkhofs, Anastasiya Starovoyt & Nicolas Verhaert
Reviewer comments:
Reviewer #2:
Point 1: At line 56, please add the prior OCT angiography (OCTA) studies for cochlea vascular imaging.
These references have been added (Line 58):
- Roberto Reif, Zhongwei Zhi, Suzan Dziennis, Alfred L. Nuttall, & Ruikang K. Wang. (2013). Changes in cochlear blood flow in mice due to loud sound exposure measured with Doppler optical microangiography and laser Doppler flowmetry. Quant Imaging Med Surg, 235–242. https://doi.org/doi: 10.3978/j.issn.2223-4292.2013.10.02
- Dziennis, S., Reif, R., Zhi, Z., Nuttall, A. L., & Wang, R. K. (2012). Effects of hypoxia on cochlear blood flow in mice evaluated using Doppler optical microangiography. Journal of Biomedical Optics, 17(10), 1060031. https://doi.org/10.1117/1.jbo.17.10.106003
- Burwood, G. W. S., Dziennis, S., Wilson, T., Foster, S., Zhang, Y., Liu, G., Yang, J., Elkins, S., & Nuttall, A. L. (2020). The mechanoelectrical transducer channel is not required for regulation of cochlear blood flow during loud sound exposure in mice. Scientific Reports, 10(1). https://doi.org/10.1038/s41598-020-66192-6
Point 2: Authors mentioned that the lateral resolution is 3.5~5 um. However, LSM04 lens used has a focal beam spot over 30 um in spec provided from the vendor, which means that the lateral resolution is more than 30 um.
Thank you for pointing this out, we erroneously used the term ‘transverse’ resolution. It actually concerns the ‘axial’ resolution. Depending on the medium used to image through, the axial resolution in our system and used system settings is either 4.1 μm in water or 5.5 μm in air. We clarified this in the manuscript (Lines 111 and 112).
We did not note the lateral resolution, however, thank you for pointing this out, we added this information in the manuscript (line 113-114). We define the lateral resolution as the beam width of the focus at 1/e², with a wavelength of 1310 nm, resulting in 20 µm for the LSM04 (https://www.thorlabs.com/images/Catalog/Imaging/2_Oct.pdf).
Point 3: For the acquired 3D OCT dataset, please explain the number of pixels/A-line, number of A-lines/B-scan, number of B-scans/C-scan, and the scanned area.
Point 4: At line 130, what is the difference between fresh and fresh-frozen?
Fresh samples have not been frozen or persevered in any way, but were as fast as possible analyzed after the death of the donor. Fresh-frozen samples, have been preserved by freezing them at a very low temperature. This helps to preserve the sample and prevent any changes or degradation that might occur if it was kept at room temperature. Fresh-frozen samples are often used when a sample needs to be stored for an extended period of time before it is used for testing or analysis. (Lines 147-150)
Point 5: Briefly, explain about the method for segmentation shown in Fig. 1A.
A c-scan was exported as JPG files, which were uploaded in Avizo (FEI Visualization Sciences Group, Thermo Fisher Scientific Inc.), together with the corresponding voxel size in each dimension. This segmentation was done manually and carefully evaluated by experts in cochlear anatomy. (Lines 124-127)
Point 6: Indicate the scale bar in Fig. 2C.
The scale bar has been added (Fig. 2C, line 224).
Point 7: At line 227, please explain why the epithelium is not seen after freezing.
We have data of one sample before and after freezing. From one sample, we cannot say for sure whether the sensory epithelium of the cochlea would degenerate after freezing, but the difference is remarkable in this one sample. The effects of freezing on samples can depend on multiple factors, including the temperature at which the sample is frozen, the length of time that it is stored, and the type of tissue being preserved, which in the cochlea involves both bony and soft tissue. Thereby, there were more fresh samples with sensory epithelium left, compared to fresh-frozen samples which did show altered sensory epithelium. Because we do not have a before and after freezing image of all these samples, we cannot be sure this effect is due to freezing, but we highly suspect that freezing causes degeneration of sensory epithelium because it was less seen in the fresh-frozen samples. (Lines 269-271) + Discussion (Lines 371-385 – visible track changes – all markup)
Point 8: At line 267, please explain how to figure out PM on the RWM in Fig. 7C, and the cause of PM generation.
In appendix B, the same sample as in Fig. 7C is illustrated. On one hand, the pseudomembrane is already visible on the microscopic image as a white-colored structure, while normally the RWM is more transparent (see microscopic images in appendix B, where you can clearly see the BM and OSL underneath the RWM). On the other hand, the PM is visible on the OCT images, as there is a thin black line visible between the RWM and PM, clearly delineating them, both indicated in the X and Y cross-sections. In Fig 7C, this thin black line between the RWM and PM, indicating the border between these membranes, is not clearly visible due to the segmentation, however, it is clearly visible in appendix B. (Lines 309-315)
The cause of PM generation is variable. The origin of a pseudomembrane is not yet known, but it might be related to middle ear infections. However, clinical evidence for this correlation is still lacking and further research is needed to investigate this. A mucosal pseudomembrane found adjacent to the surface of the RWM may impact its permeability. They may either inhibit diffusion by acting as an additional barrier, protecting the RWM. (Lines 450-455)

Round 2
Reviewer 1 Report
The reviewer appreciates the detailed revisions performed by the authors. Although, we do have three points which should still be addressed:
(1) Line 109-110: what is meant by the axial resolution of 2.62 um in water, if later the authors mention 4.1 and 5.5 um in air and water, respectively?
(2) For the variable A-scan rates, please clarify for the presented image which once were used.
(3) In Figure 4 the labeling of (A)-(D) are missing in the images. The same applies for the Supplementary figures. Maybe the authors could also add (A)-(C) labeling to Figure 3 instead of top and bottom description.
Author Response
Dear Reviewer,
Attached you can find the point-by-point response and cover letter.
Kind regards,
Lore
